# ENHANCING PERCEPTION CAPABILITIES OF MULTIMODAL LLMS WITH TRAINING-FREE FUSIONS

## ABSTRACT

Multimodal LLMs (MLLMs) equip language models with visual capabilities by aligning vision encoders with language models. Existing methods to enhance the visual perception of MLLMs often involve designing more powerful vision encoders, which requires re-aligning these vision modules with the language model, leading to expensive and time-consuming training processes. In this paper, we introduce VisionFuse, a novel integration framework that efficiently utilizes multiple vision encoders from off-the-shelf MLLMs to enhance visual perception without requiring additional training. Our approach is motivated by the observation that different MLLMs tend to focus on distinct regions of the same query and image. Moreover, we find that the feature distributions of vision encoders within an MLLM family, a group of MLLMs sharing the same pretrained LLM, are highly aligned. Building on these insights, VisionFuse enriches the visual context by concatenating the tokens generated by the vision encoders of selected MLLMs within a family. By merging the parameters of language models from different MLLMs, VisionFuse allows a single language model to align with various vision encoders, significantly reducing deployment overhead. We conduct comprehensive evaluations across multiple multimodal benchmarks using various MLLM combinations, demonstrating substantial improvements in multimodal tasks. Notably, when integrating MiniGemini-8B and SLIME-8B, VisionFuse achieves an average performance increase of over 4%.

## 1 INTRODUCTION

Multimodal LLMs (MLLMs) integrate vision encoders to Large Language Models (LLMs), allowing them to tackle multimodal tasks with emergent capabilities (Liu et al., 2024c; Lin et al., 2024; Ye et al., 2023). To handle complex and diverse multimodal tasks effectively, MLLMs require strong visual perception capabilities. A common approach to improving MLLMs' visual perception is designing better vision encoders (Cha et al., 2024; Zhang et al., 2024; Shi et al., 2024). However, these methods typically require aligning the vision encoders with the language model, which involves significant training costs due to the need for multimodal instruction fine-tuning. For instance, aligning 10 vision encoders with a 7B language model using a data-efficient MLLM pipeline requires 3,840 NVIDIA A100 GPU hours, with an estimated cost of approximately $20,000 (Yang et al., 2024). Moreover, even with substantial efforts, individual models may still exhibit limitations in their visual perception capabilities.

Recognizing these limitations, it becomes clear that different models tend to excel in distinct aspects of visual perception. For example, as illustrated in Figure 1, when asked the question, "What's the word on the right side?", the models should focus on the entire word on the coin's right side. MGM (Li et al., 2024), however, concentrates more on the lower right portion of the coin, with relatively less attention to the upper part. In contrast, SLIME (Zhang et al., 2024) directs more attention to the upper part, causing it to miss the last letter 'y'. While MGM successfully recognizes the letter 'y', its recognition of the preceding letters is suboptimal. Due to the perceptual limitations of both models, they provide incorrect answers in this text recognition task, yet demonstrate notable differences in their visual perception. This observation underscores the potential for mitigating the limitations of individual models by leveraging the complementary strengths of multiple models. By integrating the encoders in these MLLMs, our method captures a more complete target region, leading to accurate results.

Figure 1: **Different MLLMs exhibit varying visual perception capabilities.** We visualize the average cross-attention maps across all layers for two MLLMs - MGM and SLM, as well as for our method that integrates these two models, using an example to observe which areas the models focus on. It shows that our VisionFuse attention is more accurate, integrating the perceptual abilities of both MGM and SLM. Here, "MGM" represents Mini-Gemini (Li et al., 2024), and "SLM" represents SLIME (Zhang et al., 2024).

To investigate this concept, ensemble learning has been proposed as an efficient means of leveraging the capabilities of different models without requiring additional training, by simply aggregating the outputs (Jiang et al., 2023; Wan et al., 2024; Freitag et al., 2023). However, deploying multiple full models and running inference on each one introduces inefficiencies in both memory consumption and computational resources. Since vision encoders are generally smaller and less resource-intensive than entire MLLMs, a viable alternative is to integrate multiple vision encoders with a single language model. For example, Eagle (Shi et al., 2024) enhances MLLMs' visual perception by concatenating the encoded vision tokens from multiple encoders, but it necessitates a resource-intensive two-stage multimodal instruction tuning process. While this approach reduces computational overhead compared to ensemble learning, it still encounters the challenge of aligning these encoders with a single language model, which often necessitates multimodal instruction tuning and imposes considerable computational costs (Yang et al., 2024).

In this paper, we introduce VisionFuse, a novel framework designed to efficiently enhance the visual perception capabilities of MLLMs, as illustrated in Figure 2. We define the MLLM family as a group of models that share the same pre-trained language model. For example, both MiniGemini-8B (Li et al., 2024) and SLIME-8B (Zhang et al., 2024) are trained using LLaMA-3-8B-Instruct (AI@Meta, 2024), and therefore belong to the same MLLM family. We first conduct a statistical analysis of cross-attention to highlight variations in visual perception across different MLLMs, and then propose enhancing visual perception by integrating these models. Notably, we observe that vision encoders within an MLLM family exhibit similar feature distributions, making their tokens more compatible for combination. Therefore, we consider integrating these encoders to enhance perception. Furthermore, we find that merging language models within an MLLM family effectively aligns them with different vision encoders, so prior to deployment, we merge language model parameters from various MLLMs to achieve this alignment. During inference, we apply preprocessing pipelines to the input visual data (e.g., slicing the image into local patches) consistent with those used by individual MLLMs within the family, then feed the processed data into the vision encoders and projectors from each MLLM to extract visual tokens. These tokens are then concatenated to provide richer contextual information. VisionFuse effectively harnesses the visual perception capabilities of multiple multimodal models, improving performance on multimodal tasks with minimal additional inference overhead from the vision encoders.

We apply VisionFuse to different MLLM families, based on the pretrained language models including Vicuna-v1.5 (Chiang et al., 2023) and LLaMA-3-8B-instruct (AI@Meta, 2024). Compared to the individual model baselines, our approach demonstrates significant improvements across multiple multimodal datasets. Furthermore, we visualize the cross-attention to intuitively show that the model after integrating focuses on more relevant positions than the individual models.

In summary, our contributions are as follows:

- **New insights in different MLLMs**: We identify three key phenomena in various pretrained MLLMs: (1) different models focus on distinct image regions for the same input, (2) vision en-

Figure 2: Overview of VisionFuse. VisionFuse merges the language model parameters from different MLLMs within a family to align the language model with multiple vision encoders. The input image is processed through distinct preprocessing pipelines, consistent with those in MLLMs, as well as vision encoders and projectors, to extract richer visual features. These features are then concatenated with text tokens and fed into the merged language model.

coders within an MLLM family exhibit more consistent features, and (3) merging language model parameters is critical for aligning the language model with different vision encoders.

- **A training-free method to enhance the perception capabilities of MLLM**: We propose Vision-Fuse, a simple yet effective method for integrating MLLMs, based on three key observations. By merging language model parameters to align with different vision encoders and leveraging vision encoders from multiple MLLMs within a family, VisionFuse enhances visual perception and improves multimodal task performance with minimal deployment overhead. Extensive experiments demonstrate that VisionFuse effectively enhances performance on multimodal tasks. Specifically, in the integration of Mini-Gemini-8B and SLIME-8B, VisionFuse achieves a significant average improvement of over 4% across multiple multimodal benchmarks without additional training.

## 2 RELATED WORK

**Enhancing the visual perception of MLLMs.** Existing multimodal large language models (MLLMs) primarily enhance their visual perception by incorporating high-resolution inputs (Li et al., 2024), employing optimized preprocessing methods to capture richer visual features (Liu et al., 2024b; Zhang et al., 2024), and designing more effective vision modules (Zhang et al., 2024; Cha et al., 2024; Ge et al., 2024). Specifically, LLaVA-Next (Liu et al., 2024b) segments input images into local patches and uses high-quality data to train the MLLM. Mini-Gemini (Li et al., 2024) uses CLIP (Radford et al., 2021) tokens as low-resolution queries to cross-attend to another high-resolution vision encoder within co-located local windows. Honeybee (Cha et al., 2024) introduces a locality-enhanced projector that balances token management flexibility with local visual context preservation, improving both efficiency and performance in spatial understanding tasks. ConvLLaVA (Ge et al., 2024) leverages a hierarchical ConvNeXt backbone to compress high-resolution images into fewer visual tokens, enhancing efficiency while maintaining spatial understanding across diverse image resolutions. SliME (Zhang et al., 2024) refines visual adapters by employing a mixture of experts for global features and compressing local image tokens with query embeddings, improving both efficiency and performance in high-resolution tasks. Eagle (Shi et al., 2024) explores the design space of multimodal LLMs by integrating multiple vision encoders with different architectures and pretraining tasks, enhancing multimodal performance through efficient fusion strategies like direct token concatenation. However, these approaches require extensive fine-tuning to align the language model with the vision modules, leading to significant computational costs. In contrast, our work aims to improve the visual perception of MLLMs more efficiently.

**Model Merging.** Model merging seeks to consolidate multiple parameter sets into a single model without requiring retraining, providing a more memory-efficient and cost-effective alternative to model ensembling by eliminating the need to store multiple checkpoints. Existing methods for merging parameters of LLMs generally fall into two categories: merging with coefficients and parameter sparsification. Task Arithmetic (Ilharco et al., 2023) introduces task vectors to efficiently edit pre-trained models by performing arithmetic operations on weight differences, enhancing task performance without retraining. Regmean (Jin et al., 2022) computes a closed-form solution for

the merging coefficients by minimizing the difference in activation values before and after merging. SLERP (Shoemake, 1985) determines the merging coefficients for parameter addition based on the angle between the model parameters. Methods like Ties-merging (Yadav et al., 2023) and DARE (Yu et al., 2024) observe that the delta parameters of LLMs contain a significant amount of redundancy, and propose reducing conflicts between different delta parameters through pruning. However, these model merging methods are primarily designed for language or vision models to preserve the capabilities of individual models across different downstream tasks. Our method aligns language models with different vision encoders through model merging, which is significantly different from the motivations of existing model merging approaches.

## 3 MOTIVATION

In this section, we provide comprehensive visualizations and discussions from the following perspectives to clarify our motivation. The experiments are conducted using the following models: SLIME-7B (Zhang et al., 2024) and MGM-7B (Li et al., 2024), both based on Vicuna-v1.5 (Chiang et al., 2023), as well as SLIME-8B (Zhang et al., 2024) and MGM-8B (Li et al., 2024), which are based on LLaMA-3-8B-Instruct (AI@Meta, 2024). We evaluate these models on the following datasets: TextVQA (Singh et al., 2019), MME (Fu et al., 2024), and VQAv2 (Goyal et al., 2017a).

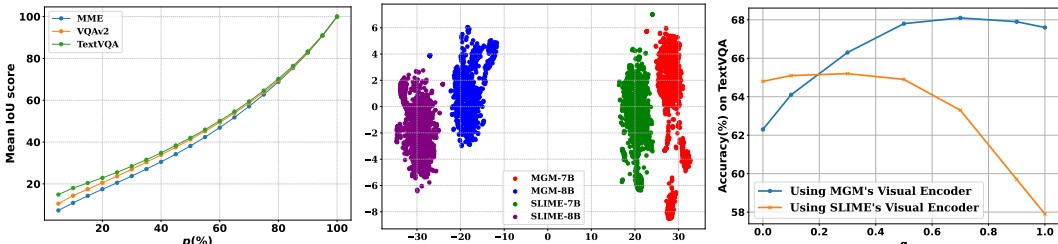

(a) IoU of top $p\%$ tokens with highest attention in different MLLMs.

(b) Visual feature distributions of four multimodal models.

(c) Accuracy on TextVQA when interpolating with different $\alpha$.

Figure 3: Summary of our exploration and observations: (a) demonstrates that different MLLMs focus on distinct image regions for the same visual and textual inputs; (b) reveals that vision encoders within an MLLM family exhibit more similar feature distributions; and (c) highlights the importance of merging language model parameters to align the language model with different vision encoders.

**Observation 1: Different MLLMs attend to different regions for the same query and visual input.** To investigate the regions of focus for different MLLMs when given the same visual input and query, we analyze SLIME-7B and MGM-7B using samples from TextVQA, MME, and VQAv2. We calculate the average attention across all layers and the average Intersection over Union (IoU) of the top $p\%$ regions (tokens) with the highest cross-attention. As shown in Figure 3a, even with the same visual input and query, different models tend to focus on distinct regions, indicating notable differences in visual perception capabilities across multimodal models. For instance, in the MME dataset, the IoU of the top 5% of tokens with the highest cross-attention score is less than 10%. As demonstrated in Figure 1, there are significant differences in the cross-attention between MGM and SLIME when presented with the same image and text query. *Thus, we propose leveraging these differences to provide the language model with richer visual information.*

**Observation 2: Visual feature distributions of encoders within an MLLM family exhibit a closer alignment.** MLLMs align visual and textual features during multimodal instruction fine-tuning, allowing visual features to serve as additional contextual information for textual input. Intuitively, vision encoders within an MLLM family should have more closely aligned visual features, as they are aligned with the same pretrained language model. To validate this hypothesis, we randomly select 100 samples from the TextVQA (Singh et al., 2019) dataset and visualize the distribution of all visual tokens from different vision encoders using t-SNE (Van der Maaten & Hinton, 2008), focusing on SLIME-7B, MGM-7B (Li et al., 2024), SLIME-8B, and MGM-8B. As illustrated in Figure 3b, the feature distributions of vision encoders within the same MLLM family are relatively similar, whereas those trained on different language models display more distinct distributions. This

indicates that *vision encoders within an MLLM family tend to generate more closely aligned visual features, making them more compatible for integration within a shared context.*

**Observation 3: Merging language model parameters aligns the language model with different vision encoders.** Previous research on model merging (Ilharco et al., 2022; Yadav et al., 2023) has primarily focused on combining delta parameters to enable a single LLM to inherit capabilities from multiple fine-tuned models. This motivate us to explore whether merging the delta parameters of the language model, relative to the pretrained language model in MLLMs, similarly facilitates alignment between the language model and different vision encoders. Using SLIME-8B and MGM-8B as examples, we calculate their delta parameters relative to LLaMA-3-8B-Instruct (Li et al., 2023b) and merge them via linear interpolation, formalized as follows:

$$\Theta_{\text{interpolate}} = \alpha \cdot (\Theta_{\text{MGM}} - \Theta_{\text{LLaMA3}}) + (1 - \alpha) \cdot (\Theta_{\text{SLIME}} - \Theta_{\text{LLaMA3}}) + \Theta_{\text{LLaMA3}}, \quad (1)$$

where $\Theta_{\text{interpolate}}$ represents the parameters after interpolation, $\Theta_{\text{MGM}}$, $\Theta_{\text{SLIME}}$, and $\Theta_{\text{LLaMA3}}$ represent the parameters of MGM-8B, SLIME-8B, and LLaMA-3-8B-Instruct, respectively. We evaluate the performance of the merged delta parameters on TextVQA. As shown in Figure 3c, when $\alpha$ is set to 0 or 1, the model corresponds to SLIME-8B or MGM-8B, respectively. However, when $\alpha$ is closer to 0.5, the model achieves optimal performance by effectively utilizing both vision encoders. This demonstrates that the delta parameters between an MLLM and its base model are the key factor enabling a single language model to align with different vision encoders. *By merging language models' delta parameters from different MLLMs within a family, we can align a single language model with multiple vision encoders.*

# 4 METHODOLOGY

Inspired by our observations, we propose VisionFuse, a simple yet effective approach for efficiently integrating different MLLMs to enhance visual perception. As illustrated in Figure 2, Vision-Fuse first merges the language models and then utilizes various vision encoders to extract richer features for the input image, which are subsequently fed into the merged LLM.

## 4.1 PRELIMINARIES

**Notation**: Let $f_{\Theta}(x)$ represents the language model of an MLLM, where $\Theta$ denotes the parameters of the language model. The input $x$ is a sequence of tokens, comprising vision tokens $V$ processed by the modality-specific encoder and text tokens $T$ generated from word embeddings. The parameters of the pre-trained language model are denoted as $\Theta_{\text{pre}}$.

**Delta parameters.** Delta parameters represent the changes in model parameters during fine-tuning (Liu et al., 2024d). In MLLMs, the delta parameters for the language model during multimodal fine-tuning are expressed as $\Theta - \Theta_{\text{pre}}$.

**Main components of the MLLM.** Existing MLLMs (Yin et al., 2023) typically consist of a modality preprocessing module that processes input data (e.g., slicing images into local patches), a modality-specific encoder that converts the data into features, a modality-specific projector that maps the encoded visual features into the text space, and an LLM that performs cross-modal reasoning by integrating these tokens with text-based input. For a given set of $M$ MLLMs, the language model parameters of the $i$-th model are denoted as $\Theta_{\text{i}}$, its preprocessing pipeline as Preprocessing$_i$, its vision encoder as Enc$_i$, and its vision projector as Proj$_i$.

## 4.2 ENSEMBLE OF DIFFERENT VISION ENCODERS

To leverage the visual perception capabilities of different MLLMs, an intuitive approach is to ensemble multiple MLLMs and aggregate their predictions. However, due to the large number of parameters in the language models, directly ensembling the entire MLLMs would result in significant computational overhead. Therefore, we explore whether it is feasible to integrate only the vision encoders, which have relatively fewer parameters, and aggregate the outputs in a way that feeds into the language model for inference.

In pre-trained MLLMs, the alignment between textual and visual inputs in the feature space allows them to be treated as a unified sequence for input into the language model. As discussed in **Obser-**

**vation 2**, visual features from MLLMs trained on the same language model are more closely aligned in the feature space. Therefore, we directly aggregate the outputs of different vision encoders within an MLLM family, treating them as distinct visual context information, which enables the language model to obtain richer visual perception. Due to the varying lengths of vision tokens from different MLLMs, we propose directly concatenating the vision tokens from these MLLMs. The process of ensembling multiple vision encoders can be formalized as follows:

$$V_i = \text{Proj}_i(\text{Enc}_i(\text{Preprocessing}_i(x))), \tag{2}$$

$$V_F = [V_1; V_2; \ldots; V_n], \tag{3}$$

where $[\cdot; \cdot]$ denotes concatenation, and $V_F$ represents the integrated visual features.

## 4.3 MERGING LLMS FROM A FAMILY OF MLLMS

A single language model cannot directly align with multiple encoders from different MLLMs, as they have not undergone alignment training. Retraining for such alignment would result in substantial computational costs. As discussed in **Observation 3**, merging language model parameters within an MLLM family helps align a language model with different vision encoders. Therefore, following the approach in (Ilharco et al., 2022), we merge these delta parameters to create a single language model capable of interpreting visual tokens from multiple vision encoders. The merging process can be formalized as follows:

$$\mathbf{\Theta}_{\text{merged}} = \mathbf{\Theta}_{\text{pre}} + \lambda \cdot \sum_{i=1}^{M} (\mathbf{\Theta}_{\text{i}} - \mathbf{\Theta}_{\text{pre}}), \tag{4}$$

where $\mathbf{\Theta}_{\text{pre}}$ represents the parameters of the shared base model, and $\lambda$ is a merging coefficient. The prediction $\hat{y}$ can then be generated as follows:

$$\hat{y} = f(V_F, T; \mathbf{\Theta}_{\text{merged}}). \tag{5}$$

---

**Algorithm 1** Procedure of Inference for VisionFuse

---

**Input:** $M$ MLLMs built upon the same pretrained language model with parameters $\mathbf{\Theta}_{\text{pre}}$, with the $i$-th model having language model's parameter $\mathbf{\Theta}_{\text{i}}$, vision encoder $\text{Enc}_i$ and preprocessing pipeline $\text{Preprocessing}_i$, input image $x$, text input $T$.
**Output:** Prediction $\hat{y}$
1: Generate merged parameters $\mathbf{\Theta}_{\text{merged}}$ using Eq. (4).          ▷ Merge language model parameters
2: **for** the $i$-th vision encoder from $1$ to $M$ **do**      ▷ Extract visual features from different encoders
3:    Extract visual features $V_i$ using Eq.( 2).
4: **end for**
5: Generate $V_F$ using Eq.( 3).
6: **return** Prediction $\hat{y}$ using Eq.( 5).

---

**Complexity analysis.** *For training complexity, our VisionFuse is significantly more cost-effective than existing methods.* Existing MLLMs often require extensive training cost to achieve alignment between the language model and vision encoders (Yang et al., 2024). Instead, our VisionFuse enhances the perception capabilities by concatenating tokens from different vision encoders and merging parameters of the language models from different MLLMs, with no additional training cost.

*For inference complexity, our VisionFuse requires some additional cost to process the longer visual tokens.* Since the computational cost of the language model of MLLMs is significantly higher than that of the vision encoder, we mainly analyze the computational cost of the language model for different methods. The Floating Point Operations (FLOPs) in language model for $i$-th MLLM are $O\left(\left(L_{\text{Enc}}^i + L_t\right)^2\right)$, where $L_{\text{Enc}}^i$ represents the length of generated tokens from $i$-th MLLM's vision encoders, $L_t$ represents the length of text tokens. Since our VisionFuse concatenates the tokens from all $M$ visual modules of MLLMs, it requires processing more tokens compared to a single MLLM. Specifically, the FLOPs of VisionFuse are $O\left(\left(\sum_{i=1}^{M} L_{\text{Enc}}^i + L_t\right)^2\right)$. To reduce the inference cost,

one can use the token pruning method to remove the redundant visual tokens (Chen et al., 2024). We analyze our method under different levels of sparsity and the results demonstrate that *our method is able to achieve better performance with lower FLOPs compared to a single model* (see Table 5).

# 5 EXPERIMENTS

**Implementation details.** We conduct experiments across various MLLM combinations, including (1) SLIME-7B (Zhang et al., 2024) and MGM-7B (Li et al., 2024) based on Vicuna-v1.5 (Chiang et al., 2023), and (2) SliME-8B (Zhang et al., 2024) and MGM-8B (Li et al., 2024) based on Llama-3-8B-Instruct (AI@Meta, 2024). We evaluate the performance of our approach on multiple multi-modal datasets, including $VQA^T$ (TextVQA) (Singh et al., 2019), MMB (MMBench) (Liu et al., 2023b), $MMB^C$ (MMBench-Chinese) (Liu et al., 2023b), MME (Fu et al., 2024), MMMU (Yue et al., 2024), VQAv2 (Goyal et al., 2017b) and Vizwiz (Gurari et al., 2018).

**Compared methods.** We compare our method with the baselines and existing leading MLLMs, including MobileVLM (Chu et al., 2023), Qwen-VL Bai et al. (2023), Qwen-VL-Chat Bai et al. (2023), IDEFICS (Laurencon et al., 2023), LLaMA-VID (Li et al., 2023b), LLaVA-1.5 (Liu et al., 2024a), VILA (Lin et al., 2024), Shika (Chen et al., 2023) and InstructBLIP (Dai et al., 2023).

Table 1: Comparison with leading methods on multimodal benchmarks. Results of VisionFuse are marked in gray. $VQA^T$: TextVQA; MMB: MMBench; $MMB^C$: MMBench-Chinese; $MMMU_{v,t}$: validation and test set of MMMU; $MME^{P,C}$: Perception and Cognition in MME. Res. indicates the resolution of the input. Percentages indicate the rate of improvement compared to the best performance of the baselines.

| Method | LLM | Res. | $VQA^T$ | VQAv2 | Vizwiz | $MME^P$ | $MME^C$ | MMB | $MMB^C$ | $MMMU_v$ | $MMMU_t$ |
|---|---|---|---|---|---|---|---|---|---|---|---|
| MobileVLM | MLLaMA 2.7B | 336 | 47.5 | - | - | 1289 | - | 59.6 | - | 26.2 | - |
| InstructBLIP | Vicuna-7B | 224 | 50.1 | - | 34.5 | - | - | - | 36.0 | - | - |
| InstructBLIP | Vicuna-13B | 336 | 50.7 | - | 33.4 | 1213 | - | - | - | 25.6 | - |
| Qwen-VL | Qwen-7B | 336 | 59.8 | 78.8 | 35.2 | - | - | 66.0 | - | - | - |
| Qwen-VL-Chat | Qwen-7B | 448 | 61.5 | 78.2 | 38.9 | 1488 | - | 68.0 | - | 35.9 | 32.2 |
| Shikra | Vicuna-13B | 336 | 52.3 | - | - | - | - | 59.2 | - | - | - |
| IDEFICS-80B | LLaMA-65B | 224 | 30.9 | - | - | - | - | 54.5 | - | - | - |
| LLaMA-VID | Vicuna-7B | 336 | - | 79.3 | 54.2 | 1521 | - | 65.1 | - | - | - |
| LLaMA-VID | Vicuna-13B | 336 | - | 80.0 | 54.3 | 1521 | - | 66.6 | - | - | - |
| LLaVA-1.5 | Vicuna-7B | 336 | 53.5 | 78.5 | 50.0 | - | - | 66.4 | 58.3 | 31.5 | - |
| LLaVA-1.5 | Vicuna-13B | 336 | 63.2 | 80.0 | 53.6 | 1531 | 295 | 65.2 | 63.6 | 36.4 | 33.1 |
| LLaVA-HD | Vicuna-13B | 336 | 62.5 | 81.8 | 57.5 | 1500 | - | 68.8 | 61.9 | - | - |
| MGM-8x7B | Mixtral-8x7B | 336 | 69.2 | - | - | 1639 | 379 | 75.6 | - | 41.8 | 37.1 |
| MGM-7B | Vicuna-7B | 336 | 65.2 | 80.4 | 52.1 | 1523 | 316 | 68.7 | 57.8 | 36.1 | 32.8 |
| SliME-7B | Vicuna-7B | 336 | 64.4 | 80.3 | 53.7 | 1544 | 383 | 68.4 | 61.3 | 37.2 | 33.4 |
| MGM-SliME-7B | Vicuna-7B | 336 | **66.9** +2.6% | **80.7** +0.4% | **54.4** +1.3% | **1563** +1.2% | **394** +2.9% | **69.6** +1.3% | **62.5** +2.0% | **37.8** +1.6% | **33.6** +0.6% |
| MGM-8B | LLaMA-3-8B-Instruct | 336 | 67.6 | 81.0 | 50.9 | 1606 | 341 | 68.1 | 62.1 | 38.2 | 36.3 |
| SliME-8B | LLaMA-3-8B-Instruct | 336 | 64.8 | 80.7 | 53.1 | 1578 | 337 | 73.2 | 69.8 | 40.8 | 37.2 |
| MGM-SliME-8B | LLaMA-3-8B-Instruct | 336 | **70.0** +3.6% | **82.1** +1.4% | **60.9** +14.7% | **1645** +2.4% | **372** +9.1% | **73.9** +1.0% | **71.9** +3.0% | **41.6** +2.0% | **38.4** +3.2% |
| MGM-8B-HD | LLaMA-3-8B-Instruct | 672 | 71.6 | 81.5 | 54.4 | 1532 | 357 | 70.6 | 65.2 | 37.0 | 36.5 |
| SliME-8B | LLaMA-3-8B-Instruct | 336 | 64.8 | 80.7 | 53.1 | 1578 | 337 | 73.2 | 69.8 | 40.8 | 37.2 |
| MGM-HD-SliME-8B | LLaMA-3-8B-Instruct | 672 | **72.7** +1.5% | **82.2** +0.9% | **55.2** +1.5% | **1600** +1.4% | **364** +2.0% | **75.1** +2.5% | **70.8** +1.4% | **41.8** +2.5% | **37.4** +0.5% |

**Main results.** We evaluate our method across several multimodal datasets and compare it against the leading MLLMs, as detailed in Table 1. Without any additional training, our approach significantly enhances performance over individual models by simply integrating vision encoders from the same MLLM family. Notably, in the integration of MGM-8B and SLIME-8B, VisionFuse incurs only a 3.4% increase in parameters due to the additional encoders employed, achieving a 4% relative improvement compared to the optimal individual model. Furthermore, the performance is on par with that of the MGM-8x7B model, which contains over six times more parameters, highlighting the parameter efficiency of VisionFuse.

**Effectiveness on different resolutions inputs.** To further assess the effectiveness of our method with varying image resolutions, we combine the high-resolution vision encoder from Mini-Gemini-HD-8B with the low-resolution vision encoder from SLIME-8B. As demonstrated in Table 1, even the low-resolution vision encoder can mitigate some of the limitations of the high-resolution encoder by providing richer visual information to the language model. The performance after integration is higher than that of either individual model. This suggests that even models utilizing high-resolution

inputs may overlook critical regions, whereas the low-resolution vision encoder can help address these gaps in visual perception.

Table 2: Ablation study on each component. We conduct ablation experiments on the merging of MGM-8B and SLIME-8B. In this context, 'V.E. of SLIME' refers to using the vision encoder from SLIME-8B, and 'V.E. of MGM' refers to using the vision encoder from MGM-8B. 'SLIME's Param' indicates the use of SLIME-8B as the language model, and 'MGM's Param' indicates the use of MGM-8B as the language model. When both 'MGM's Param' and 'SLIME's Param' are selected, we merge the parameters of MGM-8B and SLIME-8B.

| SLIME's Param | MGM's Param | V.E. of SLIME | V.E. of MGM | TextVQA | $\text{MME}^P$ | $\text{MME}^E$ | GQA | POPE |
|:---:|:---:|:---:|:---:|:---:|:---:|:---:|:---:|:---:|
| ✓ | ✓ | ✓ | ✓ | **70.0** | **1645** | **372** | **64.3** | **85.8** |
| | ✓ | | ✓ | 67.6 | 1606 | 341 | 63.2 | 85.6 |
| ✓ | | ✓ | | 64.8 | 1578 | 337 | 63.9 | 84.9 |
| | ✓ | ✓ | ✓ | 66.4 | 1485 | 294 | 62.3 | 84.2 |
| ✓ | | ✓ | ✓ | 66.9 | 1590 | 355 | 62.5 | 84.6 |
| ✓ | ✓ | ✓ | | 65.9 | 1606 | 358 | 63.1 | 83.1 |
| ✓ | ✓ | | ✓ | 68.1 | 1628 | 348 | 63.5 | 82.9 |

**Effectiveness of each component.** To assess the effectiveness of each component of our method, we perform ablation studies on several multimodal datasets, as shown in Table 2. The results demonstrate that without merging the delta parameters, the language model fails to align with the vision encoders from different MLLMs, leading to a decline in performance. On the other hand, merging the delta parameters without incorporating features from multiple vision encoders results in moderate improvements in multimodal tasks. However, these gains are limited by the absence of richer visual information. Therefore, the substantial performance enhancement is largely attributed to the inclusion of richer visual information.

**Effectiveness of integrating different vision encoders.** To further assess the effectiveness of integrating different vision encoders, we directly duplicate the vision encoder features of MGM-8B multiple times (×2, ×3, and ×4) and compare the performance of integrating MGM-8B with SLIME-8B. As shown in Figure 4, simply duplicating MGM-8B's visual tokens, despite providing the language model with more visual token inputs, does not result in performance improvement. This is because no additional visual information is introduced, further underscoring the effectiveness of integrating vision encoders from different MLLMs.

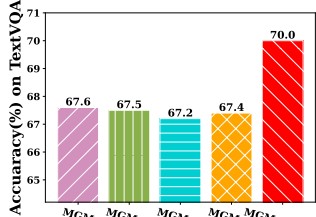

Figure 4: Comparision with directly duplicating the visual tokens of a single model.

**Influence of different merge methods.** To evaluate the impact of different model merging strategies on our approach, we apply various methods for merging language models, as presented in Table 3. Regardless of the strategy employed, our method consistently exhibits strong integration performance, surpassing that of the individual models. Among the approaches tested, Task-Arch (Ilharco et al., 2022) achieved the best results, leading us to adopt it ultimately.

Table 3: Evaluation of different model merging methods.

| Merging Method | TextVQA | $\text{MME}^P$ | $\text{MME}^E$ |
|:---|:---:|:---:|:---:|
| Task-Arch (Ilharco et al., 2022) | **70.0** | **1645** | **372** |
| Average (Choshen et al., 2022) | 70.0 | 1632 | 360 |
| Ties-Merging (Yadav et al., 2023) | 69.4 | 1626 | 352 |
| DARE (Yu et al., 2024) | 69.5 | 1632 | 352 |
| SLERP (Shoemake, 1985) | 70.0 | 1642 | 368 |

Table 4: Further evaluation using GPT. We report the average score evaluated by GPT-4o.

| Method | Average Score |
|:---|:---:|
| MGM-8B | 7.76 |
| SLIME-8B | 7.71 |
| MGM-SLIME-8B | **8.43**(+8.6%) |

**Further exploration of the enhancement of visual information richness.** Existing visual question-answering datasets primarily focus on querying specific details. To further validate that our method enables models to capture richer visual information, we construct a new dataset by randomly sampling 100 images from the COCO (Lin et al., 2014) dataset. For each image, the text prompt is: "Please describe this image in as much detail as possible." We use the GPT-4o API to score the model outputs based on the level of detail and accuracy (on a scale of 1-10), inputting

both the image and the predictions from individual models and VisionFuse. As shown in Table 4, our method, which integrates vision encoders from different multimodal models, generates more detailed and accurate image descriptions. Compared to the individual models, our approach achieves a relative improvement of 8.6%. In Figure 5, we present an example. Our method captures more comprehensive image details compared to individual models. For instance, MGM mentions onions and the location of the cutting board, while SLIME does not. Conversely, SLIME identifies spinach, which MGM overlooks. The unique visual information captured by these individual models is effectively combined when integrated using our method. Additional details of our evaluation and more example responses are provided in Appendix A.1.

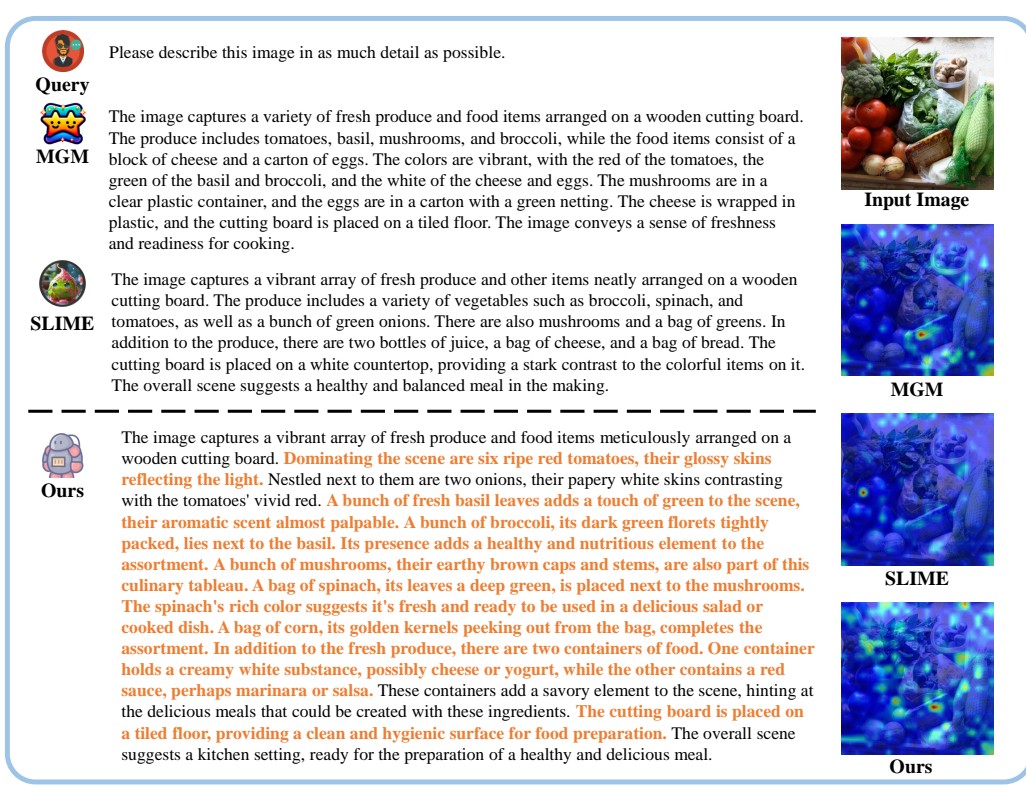

Figure 5: Qualitative comparison between individual models and our method, focusing on the richness of visual information. "MGM" refers to "MGM-8B", and "SLIME" refers to "SLIME-8B." We also visualize the average cross-attention across all layers and highlight the sections in the output of our method that provide more detailed information compared to the individual models.

**Discussions of concatenating and adding vision tokens.** The length of visual tokens in existing MLLMs varies due to differences in vision encoders, preprocessing techniques, and other factors, making direct fusion through summation challenging. To address this, VisionFuse concatenates features from different vision encoders, effectively handling inconsistent visual token lengths and providing a more flexible and adaptable aggregation method. To further explore the differences between adding and concatenating visual tokens, we conduct experiments using MGM-8B and SLIME-8B. Notably, SLIME captures additional local features through local patches, while its global features share the same length as MGM's. We employ two methods for feature summation:

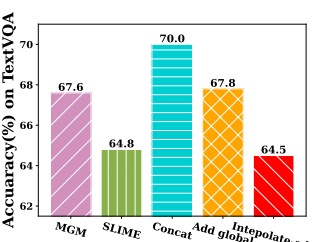

Figure 6: Comparisons of concatenating and adding.

(A) Add Global, which averages the global features from both models, and (B) Interpolate-Add, which interpolates MGM-8B's token sequence to match SLIME-8B's length before averaging. Our comparative analysis on TextVQA, shown in Figure 6, indicates that both addition methods underperform compared to concatenation, due to information loss from simplistic averaging. Developing

a more efficient strategy for fusing visual tokens across different MLLMs remains a subject for future research.

**Importance of integrating MLLMs within a family.** To assess the importance of integrating MLLMs within the same family, we evaluate the performance of integrating MLLMs across different families. Our experiments are conducted using VILA-8B, which is based on LLaMA-3-8B, and MGM-8B, which is based on LLaMA-3-8B-Instruct. Parameter merging follows Eq. 4. The parameters of LLaMA-3-8B are employed as $\Theta_{\text{pre}}$ for both VILA-8B and MGM-8B, with performance evaluated on TextVQA. As shown in Figure 7, integrating VILA-8B and MGM-8B leads to a significant performance decline, whereas combining MGM-8B with SLIME-8B results in an improvement. These findings suggest that integrating MLLMs from different families is challenging due to substantial variations in their delta parameters.

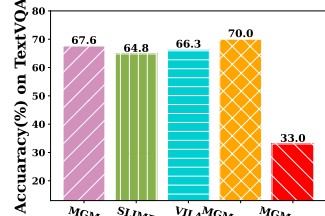

Figure 7: Performance of integrating MLLMs from different families.

**Discussion of efficiency.** As mentioned in Section 4, the increase in FLOPs for VisionFuse primarily arises from the increased length of visual tokens. To mitigate the significant rise in inference cost caused by extended visual sequences, a simple approach is to prune the token sequence. FastV (Chen et al., 2024) identifies substantial redundancy in visual tokens starting from the $3^{rd}$ layer. Inspired by this, we use the integration of MGM-8B and SLIME-8B as a case study to explore the extent of redundancy in visual tokens after integrating multiple MLLMs, and we compare the FLOPs and inference time. As shown in Table 5, after pruning 50% vision tokens starting from the $3^{rd}$ layer, the FLOPs of VisionFuse are reduced to levels below those of both MGM-8B and SLIME-8B, while maintaining superior performance compared to the individual models.

Table 5: The comparison of FLOPs and inference time under different token pruning ratios. The results of VisionFuse are marked in gray. We report the average FLOPs and inference time (second) per sample on the TextVQA and MME datasets. "Sparsity" refers to the pruning rate of vision tokens, where 0% indicates no sparsity. "Params" indicates the number of parameters. We evaluate the inference time on a single NVIDIA A800 GPU.

| Method | Sparsity | Res. | TextVQA | $\text{MME}^P$ | $\text{MME}^E$ | Params | FLOPs | Inference Time |
|---|---|---|---|---|---|---|---|---|
| MGM-8B | 0% | 336 | 67.6 | 1606 | 341 | 8.6B | 22.73T | 0.2395 |
| MGM-8B | 30% | 336 | 66.5 | 1536 | 306 | 8.6B | 9.14T | 0.2354 |
| MGM-8B | 50% | 336 | 65.5 | 1529 | 316 | 8.6B | 7.54T | 0.2292 |
| SliME-8B | 0% | 336 | 64.8 | 1578 | 337 | 8.4B | 79.04T | 0.3586 |
| SliME-8B | 30% | 336 | 64.4 | 1584 | 337 | 8.4B | 18.71T | 0.3370 |
| SliME-8B | 50% | 336 | 63.4 | 1579 | 335 | 8.4B | 14.86T | 0.2800 |
| MGM-SliME-8B | 0% | 336 | **70.0** | 1645 | **372** | 8.9B | 141.56T | 0.4171 |
| MGM-SliME-8B | 30% | 336 | 69.8 | **1649** | 368 | 8.9B | 27.18T | 0.4080 |
| MGM-SliME-8B | 50% | 336 | 69.8 | 1637 | 370 | 8.9B | 21.54T | 0.3624 |
| MGM-SliME-8B | 70% | 336 | 68.5 | 1597 | 351 | 8.9B | 16.03T | 0.3359 |

# 6 CONCLUSION AND FUTURE WORK

This paper investigates the differences in visual perception capabilities among various MLLMs and proposes a new paradigm for efficiently enhancing the perceptual abilities of MLLMs. The approach offers a simple yet effective MLLM integration strategy that requires no additional training, leveraging the distinct visual perception strengths of different MLLMs to improve performance on multimodal tasks. Overall, VisionFuse significantly enhances the perceptual abilities of individual MLLMs with minimal additional deployment overhead.

Our current analysis focuses on two MLLMs. However, when attempting to integrate more MLLMs, a direct concatenation of visual sequences results in excessively long visual token sequences, causing discrepancies that the sequence length is much longer than that in the training phase and a subsequent decline in performance. Detailed results are provided in Appendix C.1. Future work will explore methods for efficiently incorporating additional MLLMs while reducing the length of visual tokens, such as employing token merging strategies across different MLLMs or utilizing rapid fine-tuning techniques to adapt to longer input sequences.

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

# Appendix

## CONTENTS

## A   DETAILS OF EXPERIMENTAL SETTINGS

### A.1   DETAILS OF EVALUATION IN TABLE 4

For each example, we include both the image and the outputs from multiple models with the same prompt. The images are provided as URLs in the GPT-4o API. GPT-4o then returns a JSON object containing the scores for all models. The prompt is as follows:

```
Next, I will provide you with descriptions of an image generated
   by multiple models. Please evaluate these descriptions based
   on the level of detail and accuracy, and assign a score
   ranging from 1 to 10. Finally, your output only contains a
   JSON object, where each item is the model name and its
   corresponding score.

model A: ......

model B: ......

model C: ......
```

### A.2   SEARCHING DETAILS FOR THE HYPER-PARAMETERS OF MERGING METHODS

Table 6 shows the searching range of the parameters of several merging methods in Table 3.

Table 6: Searched ranges of hyperparameters.

| Method | Search Ranges of Hyperparameters |
|---|---|
| Task Arithmetic (Ilharco et al., 2022) | Scaling term: [0.1, 0.3, 0.5, 0.7, 0.9, 1.0] |
| TIES-Merging (Yadav et al., 2023) | Scaling term: [0.1, 0.3, 0.5, 0.7, 0.9, 1.0]
Ratio of retain parameters: [0.1, 0.2, 0.3] |
| DARE (Yu et al., 2024) | Scaling term: [0.1, 0.3, 0.5, 0.7, 0.9, 1.0]
Drop rate: [0.1, 0.3, 0.5, 0.7, 0.9] |

## B ADDITIONAL RELATED WORK

**Multimodal Large Language Models.** MLLMs integrate vision encoders into large language models, enabling them to handle multimodal tasks. Early models such as Flamingo (Alayrac et al., 2022) encode images and feed them into the attention layers of the language model, while Blip-2 (Li et al., 2023a) employs Q-Former to encode images into features, which are then input into the language model. Subsequent works (Liu et al., 2024c; 2023a; Lu et al., 2023; Zhang et al., 2023; Zhao et al., 2023; Zhu et al., 2023) enhance the multimodal understanding capabilities of language models through instruction fine-tuning on multimodal datasets. Further research has focused on optimizing encoder designs, extracting richer visual information, and expanding the models to handle additional modalities. For example, Eagle (Shi et al., 2024), Mini-Gemini (Li et al., 2024), SLIME (Zhang et al., 2024), and LLaVA-next (Liu et al., 2024b) introduce additional vision encoders and employ preprocessing techniques such as cropping and interpolation to handle longer visual input sequences, thereby enriching the visual information available to the language model. Honeybee (Cha et al., 2024) introduces a locality-enhanced visual projector to better bridge pre-trained vision encoders with large language models. Recent works (Lin et al., 2023; Xu et al., 2024; Han et al., 2024) also explore enabling language models to understand a wider range of modalities.

## C DISCUSSIONS AND LIMITATIONS

Although the proposed VisionFuse enhances visual perception capabilities by effectively integrating the vision encoders of different models through the concatenation of visual tokens, excessively long token sequences can introduce challenges. In Section C.1, we discuss this issue in detail. In Section C.2, we further discuss the underlying solution to alleviate this limitation.

### C.1 IMPACT OF THE TOKEN SEQUENCE LENGTH

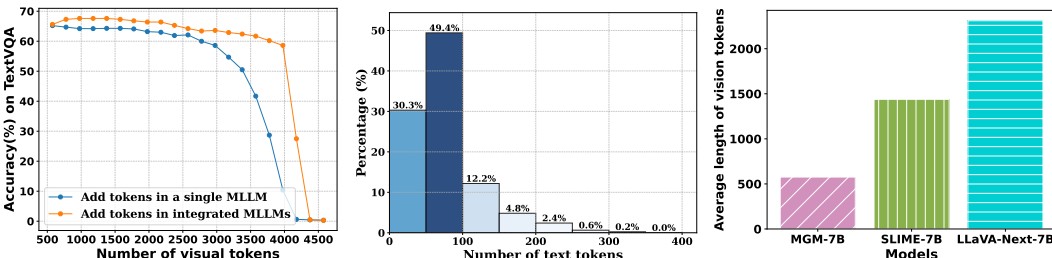

(a) Impact of increasing token length on model performance.

(b) Statistics of text token length in TextVQA.

(c) Average length of vision tokens in TextVQA.

Figure 8: Exploration of the impact of sequence length: (a) demonstrates the significant performance degradation caused by visual sequences that are excessively long and inconsistent with the training phase; (b) provides statistics on text sequence lengths in the TextVQA dataset; and (c) presents the average length of visual sequences during testing on TextVQA for the three models.

In this part, we investigate the impact of the token sequence length on the model performance. We progressively increase the number of random visual tokens and show the results of MGM-7B (a single MLLM) and our VisionFuse (integration of MGM-7B and SLIME-7B). As shown in Figure 8a, a significant decline in model performance is observed when the length of visual tokens exceeds a certain threshold. This decline can be attributed to the mismatch between the shorter token lengths used during training and the longer ones encounter during inference.

To verify the above conclusion, we also analyze the number of input text and visual tokens for MGM-7B during training and inference on the TextVQA dataset. As illustrated in Figure 8b, the majority of samples contain between 0 and 100 text tokens. During training, the visual token sequence length for both models is capped at 4096, meaning that the models are not exposed to sequences longer than this during multimodal fine-tuning. This limitation leeds to degraded performance when longer sequences are encountered during testing. As shown in Figure 8c, the total visual token length for MGM-7B, SLIME-7B, and LLaVA-Next-7B exceeds 4096, which surpasses the sequence

length encounters during training. Consequently, all three MLLMs may not learn how to handle the longer token sequences, resulting in a marked performance decline when encountering these token sequences after the integration of these MLLMs, as illustrated in Table 7.

## C.2 EXPLORATION OF INTEGRATING MORE MODELS THROUGH TOKEN PRUNING

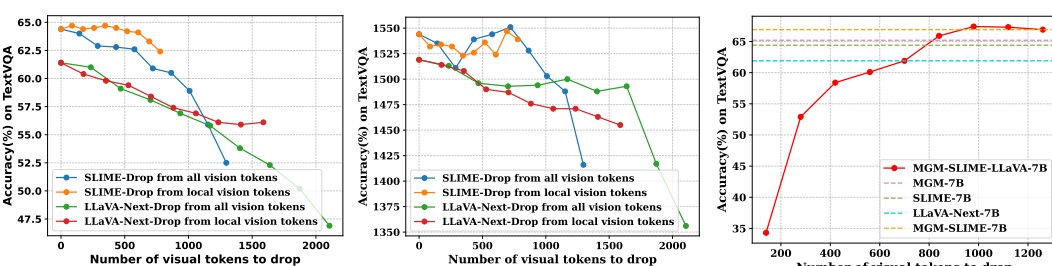

(a) Performance under different dropping numbers on TextVQA.

(b) Performance under different dropping numbers on MME.

(c) Performance under different dropping numbers of integration.

Figure 9: Exploration of redundancy in visual tokens: (a) and (b) examine the redundancy of visual tokens in SLIME-7B and LLaVA-Next-7B across the TextVQA and MME datasets. (c) investigates the effect of randomly dropping tokens on performance after integrating these two models.

To address the issue of inconsistency between training and testing caused by excessively long input sequences, one straightforward solution is to fine-tune the model with longer visual input sequences. As shown in Table 7, the 8B models, having been trained on longer token sequences, demonstrate significant performance gains with the direct integration of the three MLLMs. However, training on longer sequences comes with considerable computational costs. To mitigate this challenge more efficiently, we investigate the possibility of directly reducing the token sequence length before LLM's inference. Previous work on token pruning suggests that visual features in MLLMs exhibit substantial redundancy (Bolya et al., 2022; Chen et al., 2024), indicating that pruning redundant tokens could be a feasible solution. In the following, we assess the effectiveness of token pruning in mitigating the negative impact of inconsistent sequence lengths between training and testing, using the integration of MGM-7B, LLaVA-Next-7B, and SLIME-7B as case studies.

It is worth noting that both SLIME-7B and LLaVA-Next-7B enhance visual representations by incorporating a large number of local features through additional augmentations, resulting in significantly longer visual sequences compared to MGM-7B. We therefore assess the impact of randomly dropping visual tokens on the performance of these two models and explore the extent of redundancy in their local features. Specifically, we test two strategies on the TextVQA and MME datasets: (1) randomly dropping a specified number of tokens from the entire set of visual tokens, and (2) randomly dropping a specified number of tokens exclusively from the local features. As shown in Figures 9a and 9b, randomly dropping SLIME-7B's visual tokens leads to a substantial performance decrease, whereas removing fewer than 500 visual tokens exclusively from the local features does not result in a significant performance loss. This indicates that SLIME-7B's local features exhibit considerable redundancy. In contrast, random token pruning from either the full token set or just the local features in LLaVA-Next-7B results in a notable performance decline. Based on these observations, we recommend pruning more visual tokens in SLIME-7B while limiting the number of tokens drop in LLaVA-Next-7B.

We implement a strategy of randomly dropping visual tokens to reduce the token sequence length, enabling the language model to accommodate a larger number of visual tokens from different models. We first conduct experiments on the TextVQA dataset by integrating the MGM-7B, SLIME-7B, and LLaVA-Next-7B models. As shown in Figure 9c, performance improves when a certain number of tokens are dropped, after which further pruning may lead to a decline in performance. Ultimately, when approximately 1000 tokens are dropped, the performance of the three-model integration surpasses that of the two-model integration. Additionally, as presented in Table 7, random pruning of 1000 tokens leads to a notable improvement in overall performance. However, different MLLMs exhibit varying degrees of token sparsity, making it inconvenient to test each one before integration. We leave the design of a more efficient cross-model token fusion/pruning strategy for future work.

Table 7: Comparison with leading methods on multimodal benchmarks. Our results are marked in gray. $\text{VQA}^T$: TextVQA; MMB: MMBench; $\text{MMMU}_{v,t}$: validation and test set of MMMU; $\text{MME}^{P,C}$: Perception and Cognition in MME. Res. indicates the resolution of the input image. Percentages indicate the rate of improvement compared to the best performance of the baseline. * denotes randomly dropping 1000 vision tokens.

| Method | LLM | Res. | $\text{VQA}^T$ | $\text{MME}^P$ | $\text{MME}^E$ | MMB | $\text{MMMU}_v$ | $\text{MMMU}_t$ |
|---|---|---|---|---|---|---|---|---|
| MobileVLMI | MLLaMA 2.7B | 336 | 47.5 | 1289 | - | 59.6 | 26.2 | - |
| InstructBLIP | Vicuna-7B | 224 | 50.1 | - | - | 36.0 | - | - |
| InstructBLIP | Vicuna-13B | 336 | 50.7 | 1213 | - | - | 25.6 | - |
| Qwen-VL | Qwen-7B | 336 | 59.8 | - | - | 66.0 | - | - |
| Qwen-VL-Chat | Qwen-7B | 448 | 61.5 | 1488 | - | 68.0 | 35.9 | 32.2 |
| Shikra | Vicuna-13B | 336 | 52.3 | - | - | 59.2 | - | - |
| IDEFICS-80B | LLaMA-65B | 224 | 30.9 | - | - | 54.5 | - | - |
| LLaMA-VID | Vicuna-7B | 336 | - | 1521 | - | - | - | - |
| LLaMA-VID | Vicuna-13B | 336 | - | 1521 | - | 66.6 | - | - |
| LLaVA-1.5 | Vicuna-7B | 336 | 53.5 | - | - | 66.4 | 31.5 | - |
| LLaVA-1.5 | Vicuna-13B | 336 | 63.2 | 1531 | 295 | 65.2 | 36.4 | 33.1 |
| LLaVA-HD | Vicuna-13B | 336 | 62.5 | 1500 | - | 68.8 | - | - |
| MGM-8x7B | Mixtral-8x7B | 336 | 69.2 | 1639 | 379 | 75.6 | 41.8 | 37.1 |
| MGM-7B | Vicuna-7B | 336 | 65.2 | 1523 | 316 | 68.7 | 36.1 | 32.8 |
| SliME-7B | Vicuna-7B | 336 | 64.4 | 1544 | 383 | 68.4 | 37.2 | 33.4 |
| LLaVA-Next-7B | Vicuna-7B | 336 | 61.9 | 1519 | 334 | 65.6 | 30.7 | 30.5 |
| MGM-SliME-7B | Vicuna-7B | 336 | **66.9** (+2.6%) | **1563** (+1.2%) | **394** (+2.8%) | **69.6** (+1.3%) | **37.8** (+1.6%) | **33.6** (+0.6%) |
| MGM-SliME-LLaVA-7B | Vicuna-7B | 336 | 20.1 (-69.2%) | 1220 (-21.0%) | 268 (-30.2%) | 60.7 (-13.2%) | 29.9 (-20.9%) | 21.2 (-38.0%) |
| MGM-SliME-LLaVA-7B* | Vicuna-7B | 336 | **67.4** (+3.4%) | **1570** (+1.7%) | **397** (+3.7%) | **70.2** (+2.2%) | **37.8** (+1.6%) | **33.9** (+1.5%) |
| MGM-8B | LLaMA-3-8B-Instruct | 336 | 67.6 | 1606 | 341 | 68.1 | 38.2 | 36.3 |
| SliME-8B | LLaMA-3-8B-Instruct | 336 | 64.8 | 1578 | 337 | 73.2 | 40.8 | 37.2 |
| LLaVA-Next-8B | LLaMA-3-8B-Instruct | 336 | 64.6 | 1604 | 318 | 72.1 | 40.7 | 37.0 |
| MGM-SliME-8B | LLaMA-3-8B-Instruct | 336 | **70.0** (+3.6%) | **1645** (+2.4%) | **372** (+8.3%) | **73.9** (+1.0%) | **41.6** (+2.0%) | **38.4** (+3.2%) |
| MGM-SliME-LLaVA-8B | LLaMA-3-8B-Instruct | 336 | **70.9** (+4.9%) | **1660** (+3.4%) | **369** (+8.2%) | **75.1** (+2.6%) | **41.7** (+2.2%) | **38.7** (+4.0%) |

## D PERFORMANCE ON INTEGRATION OF MLLMs FROM DIFFERENT FAMILIES

To further evaluate the performance of VisionFuse on MLLMs from different families, we conduct experiments with MGM-7B, MGM-8B (Li et al., 2024), VILA-7B, and VILA-8B (Lin et al., 2024), as shown in Table 8. Vicuna-v1.5 is trained on LLaMA-2-7B, and LLaMA-3-8B-Instruct shares the same architecture as LLaMA-3-8B, meaning they have the same structure but different parameters. For merging the language model of MGM-7B and VILA-7B, we compute the delta parameters based on LLaMA-2-7B, and for merging the language model of VILA-8B and MGM-8B, we use delta parameters based on LLaMA-3-8B. Notably, the delta parameters differences between MGM-7B and VILA-7B are larger compared to those between MGM-7B and SLIME-7B, and similarly, the delta parameters differences between MGM-8B and VILA-8B are larger compared to those between MGM-8B and SLIME-8B. While improvements are achieved on some datasets, there is a significant performance drop on others. This is attributed to the substantial differences in delta parameters, resulting in a significant alignment error between the merged language model and the individual vision encoders.

Table 8: Merging multimodal large language models from different families.

| Method | LLM | TextVQA | $\text{MME}^P$ | $\text{MME}^E$ | GQA | POPE | $\text{MMMU}_v$ |
|---|---|---|---|---|---|---|---|
| MGM-7B | Vicuna-v1.5 | 65.2 | 1523 | **316** | **64.5** | 84.1 | **36.1** |
| VILA-7B | LLaMA-2-7B | 64.4 | 1533 | 293 | 62.3 | **85.5** | 35.2 |
| MGM-VILA-7B | LLaMA-2-7B | **68.6** (+5.2%) | **1534** (+0.0%) | 298 (-5.7%) | 63.4 (-1.7%) | 84.9 (-0.7%) | 35.3 (-2.2%) |
| MGM-8B | LLaMA-3-8B-Instruct | **67.6** | **1606** | 341 | **64.3** | **85.8** | **38.2** |
| VILA-8B | LLaMA-3-8B | 66.3 | 1577 | 326 | 61.9 | 84.4 | 36.9 |
| MGM-VILA-8B | LLaMA-3-8B | 33.0 (-51.2%) | 1496 (-6.8%) | **349** (+2.3%) | 50.3 (-21.8%) | 83.9 (-2.2%) | 33.2 (-13.1%) |

## E MORE VISUALIZATIONS

We provide additional samples and visualizations to further demonstrate the effectiveness of our method. We visualize the average cross-attention across all layers of the model before and after integration. 'MGM' refers to MGM-8B, and 'SLIME' refers to SLIME-8B.

In Figure 10, it can be observed that after integrating multiple MLLMs using our method, the model captures more information and attends to a wider range of target regions. For example, in the left sample, the task is to count the number of pictures on the wall. Each model fails to detect all the pictures, resulting in an incorrect answer. However, after integrating multiple MLLMs using VisionFuse, the model successfully attends to all the pictures and provides an accurate answer. In the

right sample, individual models tend to focus on only a portion of the sign, leading to incomplete answers. In contrast, VisionFuse enables the model to attend to a more comprehensive region, resulting in a more accurate response.

In Figure 11, we visualize two examples from the TextVQA dataset. The target texts in these examples are extremely small, requiring the model to possess strong fine-grained perception capabilities for accurate recognition. While both MGM-8B and SLIME-8B identify the target regions, neither is able to provide the correct answers. In contrast, our method, which concatenates visual tokens from both models, significantly enhances the model's fine-grained perception, enabling it to correctly identify the results.

In Figure 12, we present two examples from the VQAv2 dataset. Although both MGM-8B and SLIME-8B successfully attend to the target regions, neither could correctly identify the color of the target object. Interestingly, the colors perceived by the two models are complementary to the actual target colors. After integrating the two models, our method is able to accurately recognize the correct color.

In Figure 13, MGM focuses on the overall layout of the room, mentioning the white appliances in the kitchenette and the presence of a child near the door, while SLIME emphasizes the cozy atmosphere of the kitchen and living room, describing the natural light and hinting at additional rooms beyond the door. Our method, however, not only retains the overall layout but also highlights key details, such as the red ketchup bottle, the blue mustard bottle, and the presence of a woman and child, adding life and richness to the scene.

In Figure 14, MGM describes the kitchen, noting the blender, the soup's greenish-yellow color, and objects like a cutting board and a plastic bag in the background. SLIME, on the other hand, focuses on vibrant details, mentioning objects like a red pepper, a green bottle, and a white bowl. Our method captures all these elements while adding further details, such as the smoothie being green—suggesting it may be made from leafy greens—and the contrast between the wooden countertop and the white blender, enriching the contextual understanding of the scene.

In Figure 15, MGM highlights the modern architecture of the train station, focusing on the glass roof that lets in natural light and the presence of travelers with luggage, reflecting a busy travel period. SLIME provides more specificity by identifying the station as Saint Pancras in London and emphasizing the constant flow of passengers. Our method combines these insights and adds further details, such as the gray tiles reflecting light, the French sign directing passengers to the exit, and the two visible trains in the background, one white and one blue, adding depth and vibrancy to the depiction of the station.

In Figure 16, MGM describes the cozy living room, focusing on the reddish-brown walls, the arrangement of the furniture, and the natural lighting that creates a homely atmosphere. SLIME shifts the focus slightly, noting a gray couch, a vibrant red wall, and a cat lounging on the couch, while also emphasizing the warmth from the natural light. Our method integrates these elements and adds more specific details, such as the orange wall complementing the white ceiling, the remote control on the coffee table, suggesting an upcoming TV session, and the cat perches on the left side of the couch. This combined description provides a more comprehensive and balanced portrayal of the room's inviting and comfortable ambiance.

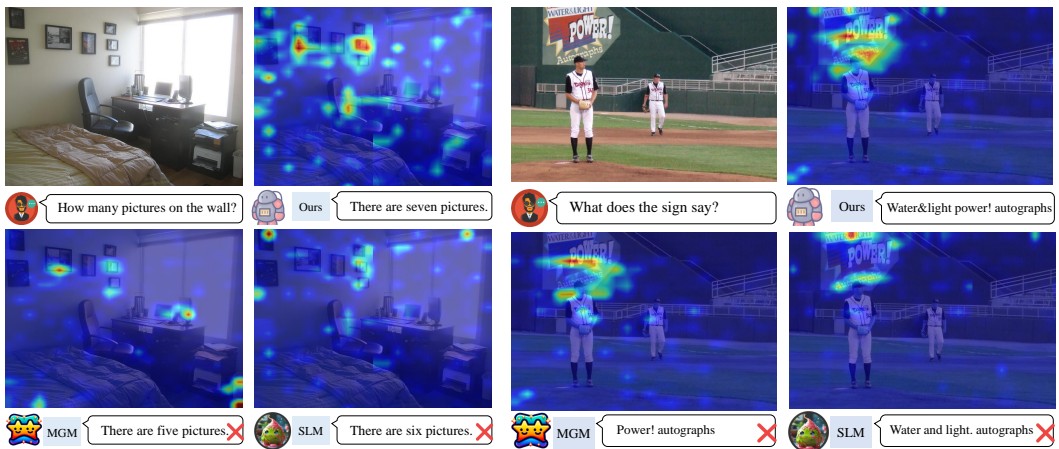

Figure 10: Enhancement of the perception regions.

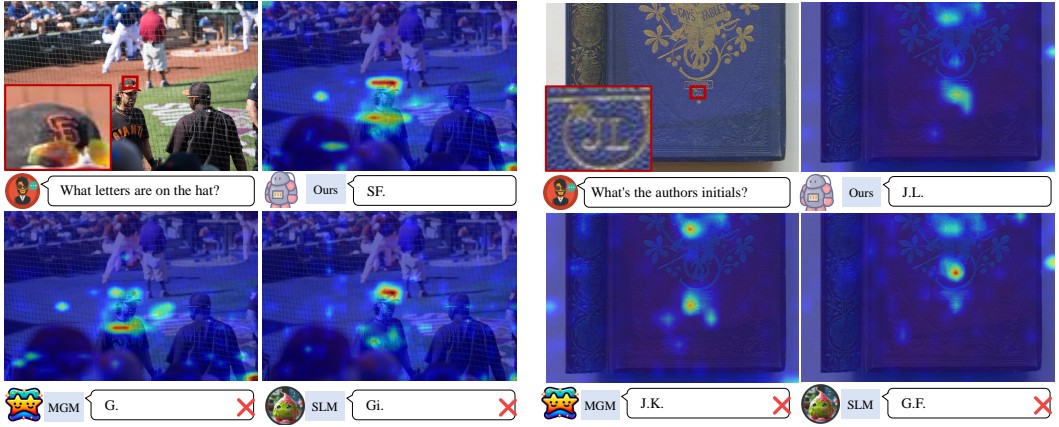

Figure 11: Enhancement of fine-grained perception

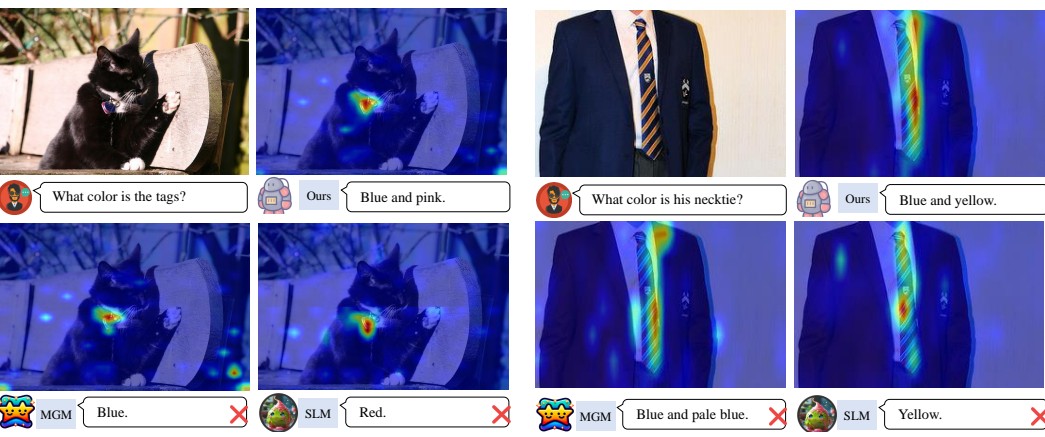

Figure 12: Enhancement of colors perception

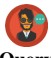

**Query** Please describe this image in as much detail as possible.

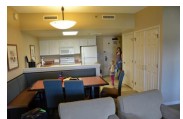
**Input Image**

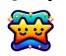

**MGM** The image is a photograph of a hotel room that includes a kitchenette and a living area. The kitchenette is equipped with white appliances, including a refrigerator, microwave, and stove. The countertops are a light color, and the cabinets are white. The living area features a beige sofa and a coffee table with a glass top. There is a dining table with four chairs, and a child is standing near the door, possibly exiting the room. The room is well-lit, with natural light coming from the ceiling fixture and the door. The walls are painted in a light color, and there are no visible texts or distinctive brands in the image. The style of the room suggests a functional and comfortable space designed for travelers.

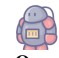

**SLIME** The image captures a cozy and inviting living space. Dominating the scene is a kitchen, characterized by white cabinets and a white refrigerator. The kitchen is equipped with a sink and a stove, ready for culinary adventures. Adjacent to the kitchen, a dining area unfolds with a wooden table and chairs, offering a space for meals and conversations. A comfortable couch and a chair are positioned in the living room, providing a spot for relaxation. The room is bathed in natural light, streaming in from a window. A door leads to a hallway, hinting at more rooms beyond. The overall atmosphere is one of warmth and homeliness.

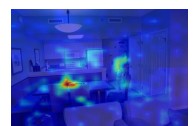
**MGM**

**Ours** The image captures a cozy and inviting hotel room. Dominating the space is a kitchenette, equipped with a white refrigerator, a white stove, and a white sink. **The kitchenette is neatly arranged, with the refrigerator on the left, the stove in the middle, and the sink on the right.** Adjacent to the kitchenette is a dining area, featuring a wooden table accompanied by four chairs. **The table is set with a red bottle of ketchup and a blue bottle of mustard, ready for use.** The living area is furnished with a beige sofa and two matching armchairs, providing a comfortable space for relaxation. A wooden coffee table sits in the center of the living area, offering a surface for placing items. **A woman and a child are present in the room, adding a touch of life to the scene. The woman is standing near the kitchenette, while the child is walking towards the door.** Their presence suggests that the room is currently in use. **The walls of the room are painted in a soothing light beige color, complementing the beige carpet underfoot.** The overall layout and decor of the room suggest a well-maintained and welcoming environment.

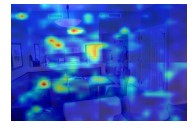
**SLIME**

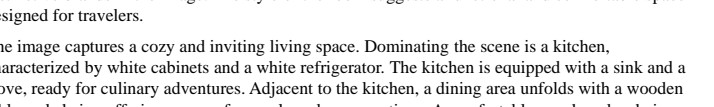
**Ours**

Figure 13: A sample for detailed descriptions.

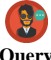

**Query** Please describe this image in as much detail as possible.

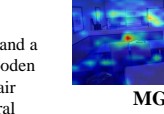
**Input Image**

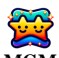

**MGM** The image shows a blender filled with a greenish-yellow liquid, which appears to be a pureed soup or sauce. The blender is placed on a kitchen counter, and there are various kitchen items in the background, including a cutting board with tomatoes and a plastic bag. The blender is a white Oster brand, and the lid is on top. The background suggests a home kitchen setting with a focus on food preparation.

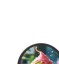

**SLIME** The image captures a moment in a kitchen where a blender, filled with a vibrant yellow soup, is placed on a wooden countertop. The blender, with its white lid and base, stands out against the backdrop of the kitchen. The countertop is not just a stage for the blender, but also a home to other kitchen items. A red pepper, a green bottle, and a white bowl are scattered around the blender, adding a splash of color to the scene. The perspective of the image is from the side of the blender, giving a clear view of the soup inside and the surrounding kitchen items. The image does not contain any discernible text. The relative positions of the objects suggest a well-used and functional kitchen space.

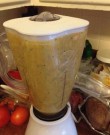
**MGM**

**Ours** The image captures a moment in a kitchen where a blender, filled with a green smoothie, is placed on a wooden countertop. The blender, which is white and has a blue label on the front, is the main focus of the image. **The smoothie inside the blender is a vibrant green color, suggesting it might be made from leafy greens or other green ingredients. The countertop on which the blender sits is made of wood and provides a warm contrast to the cool white of the blender. In the background, there are various kitchen items scattered around, including a cutting board, a knife, and a bowl.** These items suggest that the kitchen is in use and that someone might have been preparing a meal or a snack. The image does not contain any text or discernible actions. The relative positions of the objects suggest a typical kitchen scene with the blender in the foreground and the other items in the background. The image does not provide any information about the time of day or the specific location within the kitchen.

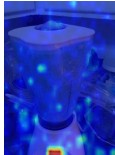
**SLIME**

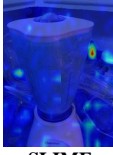
**Ours**

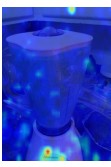

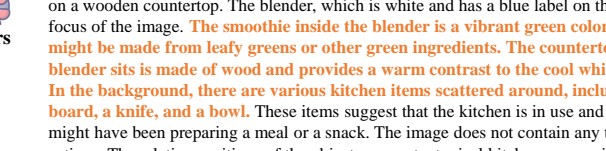

Figure 14: A sample for detailed descriptions.

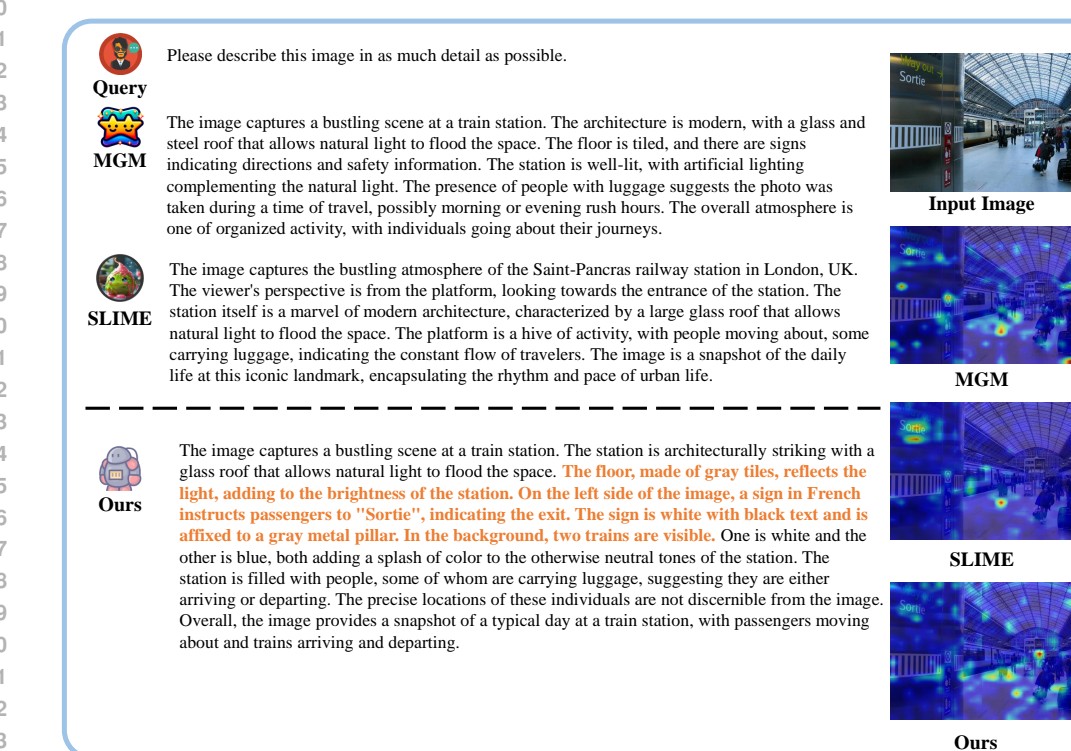

Figure 15: A sample for detailed descriptions.

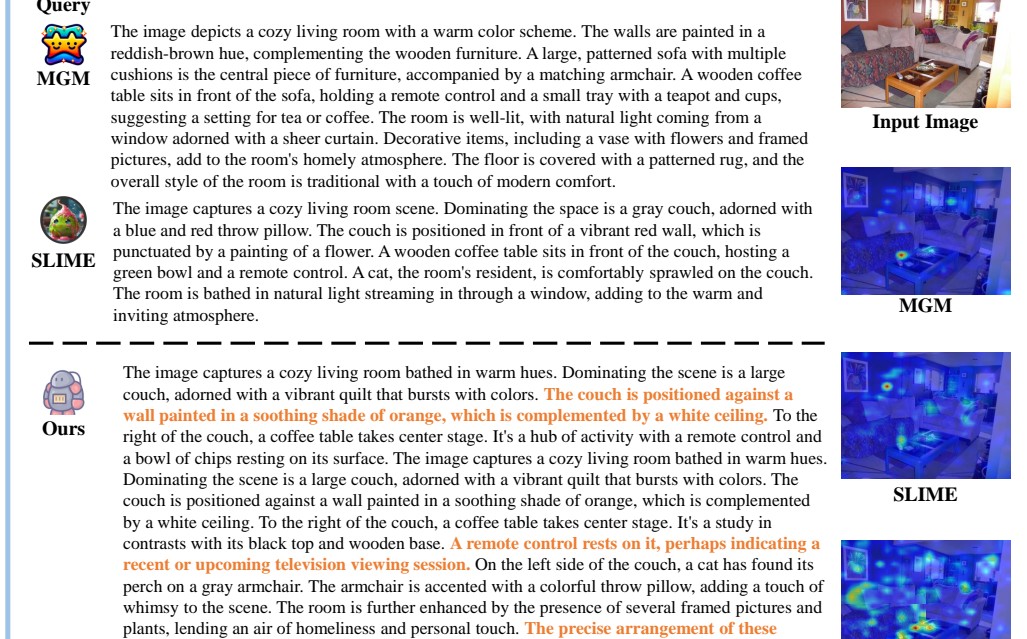

Figure 16: A sample for detailed descriptions.

