# OpenReview forum: "Enhancing Perception Capabilities of Multimodal LLMs with Training-Free Fusions"
_ICLR.cc/2025/Conference — ICLR 2025 Conference Withdrawn Submission_

### Official Review · Reviewer_1yME · 2024-10-30

**Soundness:** 2
**Presentation:** 3
**Contribution:** 2
**Rating:** 5
**Confidence:** 5

**Summary:**

This paper proposes a training-free model ensemble method for multimodal LLMs by concatenating the vision tokens and merging the LLM weights. Through exploratory experiments, the paper makes several observations about MLLMs: 1) different MLLMs focus on different image regions; 2) Vision features from MLLMs from the same base LLM exhibit similar distribution; 3) Merging LLM weights is critical to combine vision tokens from different MLLMs. Based on these observations, the paper further proposes the VisionFuse method which combines vision tokens from different MLLMs by concatenation and merging LLM's weights. Experiments of combining MGM and SliME show the effectiveness of the proposed method.

**Strengths:**

1. The general idea of combining multiple MLLMs with limited additional cost is meaningful.
2. The observations and discussions could provide some insights for the community.
3. The overall writing is clear and easy to follow.

**Weaknesses:**

1. The overall positioning of the paper is not very proper. The paper positions itself as combining different vision encoders, so I expect to see combining different types of vision encoders (e.g. CLIP, Siglip, DINOv2 ...) which are shown to be effective in Eagle, BRAVE, and Cambrian-1 by providing more complementary vision information. However, the overall method is more like a general MLLM merging method. The gain of the proposed method comes from different aspects: different vision information due to different preprocess and compression methods; LLM ensemble (different training data & training randomness); and different attention patterns from LLM to vision tokens.
2. The generalization ability of the proposed method is not well verified, and experiments are mainly conducted based on MGM+SliME. The paper should include more experiments with different MLLM combinations and different numbers of MLLMS in each combination to show the effectiveness of the proposed method.
3. The paper claims that one big advantage of the proposed method is that it does not require training. However, this relies on the assumption that you already have proper MLLMs with the same base LLM and distinct training data + vision processing structures. However, methods like Eagle only need to train the MLLM with different vision encoders once.
4. One major disadvantage of the proposed method is the additional token length, which is especially severe when considering combining multiple MLLMs more than 2 or MLLMs with long token lengths. The token pruning method used as a mitigation approach is still not optimal and might hurt the performance in certain tasks (e.g. DocVQA, InfoVQA, ChartQA) or with MLLMs that already have compressed vision tokens with little redundancy.

**Questions:**

1. What is the reason for choosing MGM+SliME for most of the experiments instead of using simpler models like LLaVA-1.5?
2. For the MGM-VILA-7B in Table-8, why do the performances on TextVQA and MME-p increase while others decrease?

---

### Official Review · Reviewer_QtQy · 2024-11-03

**Soundness:** 3
**Presentation:** 3
**Contribution:** 2
**Rating:** 5
**Confidence:** 5

**Summary:**

The paper introduces VisionFuse, a novel training-free method that efficiently utilizes multiple vision encoders from off-the-shelf MLLMs to enhance visual perception. It offers some intriguing insights. For instance, even when given the same query and image, different MLLMs focus on distinct regions. Furthermore, the authors discover that the feature distributions of vision encoders within an MLLM family are highly aligned. Leveraging these insights, they merge the parameters of language models from various MLLMs, enabling a single language model to effortlessly align with multiple vision encoders. Consequently, the proposed method achieves an average performance increase of over 4% when integrating MiniGemini-8B and SLIME-8B. Overall, the proposed VisionFuse method demonstrates the efficiency of merging parameters from multiple MLLMs, thereby harnessing the strengths of various different encoders in a unified approach.

**Strengths:**

1.	The paper introduces a training-free method called VisionFuse, designed to enhance the perception capabilities of MLLMs. This approach enables the utilization of multiple vision encoders from various MLLMs by merging the parameters of their language models. Experiments demonstrate that this method achieves a notable average improvement of over 4% across multiple benchmarks.
2.	The article presents three intriguing insights that could inspire researchers in the development of MLLMs. The visualizations and discussions provided are comprehensive and insightful.
3.	The significance of this work is good. By demonstrating a practical and efficient approach to integrating diverse vision encoders from various MLLMs into a cohesive framework through the merging of language model parameters, the paper not only advances MLLMs in multimodal applications but also enriches the broader field of vision-language integration with valuable insights.

**Weaknesses:**

1.	The authors propose that integrating a Multimodal Encoder with VisionFuse enhances the capabilities of MLLMs, as indicated in Equation (4), which suggests the potential to handle more than two MLLMs. However, the primary experiments focus on the integration of two MLLMs, such as MGM-8B and SLIME-8B. Therefore, the question arises: when integrating more than two MLLMs, how should the fusion coefficients be balanced and optimized to ensure effective integration?
2.	Does the paper discuss methods that can support the integration of scaled-up models, such as 65B or 72B models?
3.	From Figure 3, it appears that the enhancement through mixed parameter fusion is biased towards the selection of visual encoders. If an unsuitable visual encoder is used, it seems that performance could plummet. Are there any guidelines in practical applications for selecting the appropriate MLLMs for fusion enhancement without causing a decline in performance?

**Questions:**

The questions raised in this section are the same as the weaknesses outlined above.

---

### Official Review · Reviewer_dCeM · 2024-11-04

**Soundness:** 2
**Presentation:** 2
**Contribution:** 2
**Rating:** 5
**Confidence:** 4

**Summary:**

This paper introduces VisionFuse, an integration framework designed to enhance the visual perception capabilities of multimodal large language models (MLLMs) by merging different models from the same model family. The approach is built on three key observations: (i) Different MLLMs focus on varying regions of the same visual input for the same query; (ii) The visual feature distributions of encoders within an MLLM family show closer alignment; and (iii) Merging language model parameters helps align the language model with different vision encoders. The authors evaluate VisionFuse on the MGM and SliME models, demonstrating a significant improvement achieved by merging the two models.

**Strengths:**

- VisionFuse is a training-free method that can be directly applied to different models within the same MLLM family.
- The evaluation results in Table 1 demonstrate the effectiveness of the VisionFuse method.
- The authors also perform extensive experiments and ablation studies to further assess the method's effectiveness.

**Weaknesses:**

- As shown in Table 1, the proposed method’s improvements on stronger MLLMs are more limited compared to smaller models. This suggests that the method may not perform as effectively on top-tier models. Additionally, the largest model evaluated in Table 1 is only 8B, which is relatively small compared to current state-of-the-art MLLMs. It would be beneficial for the authors to test the method on larger models with top-tier performance (such as LLaVA-OneVision-Qwen2-72B, LLaVA-NeXTVideo-Qwen2-72B, and Qwen2-VL 72B), as this would help demonstrate the scalability of the proposed approach.
- The benchmarks chosen in this paper are mostly from general domains and are somewhat outdated. More recent and vision-centric benchmarks, as well as new MLLM benchmarks, are now available. These newer, more challenging benchmarks would better reflect the true capabilities of the proposed method.

**Questions:**

- After fusion, the resulting MLLM processes much longer visual inputs compared to the base models, as it concatenates vision features from multiple vision encoders into a single sequence. A relevant question arises: what if, instead of fusion, we simply increase the length of visual tokens in the base model (e.g., by expanding the input resolution)?

---

### Official Review · Reviewer_nX7t · 2024-11-12

**Soundness:** 3
**Presentation:** 3
**Contribution:** 2
**Rating:** 5
**Confidence:** 4

**Summary:**

This work proposes a new method: "VisionFuse" that ensembles different MLLMs by concatenating vision tokens and delta parameters of LLMs.

**Strengths:**

- Makes 3 significant observations on VLM ensembles: multi-encoder pays attention different complimentary regions of the image, visual embeddings of the vision encoders are better aligned when trained with same LLM , delta parameter merging of different LLM's help leverage different vision encoders from different MLLM family. These observations help them devise VisionFuse method.
- Training-Free Fusion Approach: VisionFuse addresses a critical need to enhance MLLMs’ perceptual abilities without incurring additional training costs. This "training-free" feature is a significant contribution that helps plugging in diverse models during deployment.
- The authors conduct exhaustive experiments including inference time/ accuracy to show the effectiveness of multi-encoder, LLM ensemble, token pruning (which model to prune more from) and ablations on the LLM ensemble techniques.

**Weaknesses:**

- While the paper shows exhaustive experiments on combining two MLLM families : SLIME and MGM, it is unclear how the method will scale with more than 2 MLLM due to complexity of vision token length as noted in paper. Especially, as shown in Table 8, the VisionFuse will not work when there is huge difference in the delta parameters of the LLMs. This limits the scope of this method to generalize to different MLLMs. Can the authors propose planned solutions to make the fusion more robust for any MLLM ?
- Novelty: In terms of novelty of the fusion method : the vision token concatenation was from [1], and the delta parameter integration for LLM is from [2]. Hence, the paper does not technically contribute towards the fusion methodology/ algorithm itself.
- In fig.4, there is an ablation to show the importance of complimentary features of encoders. It is unclear how to choose encoders that have complimentary features ?



1. Eagle: Exploring the design space for multimodal llms with mixture of encoders. arXiv preprint arXiv:2408.15998, 2024.
2. Editing models with task arithmetic. In ICLR. OpenReview.net, 2023. URL

**Questions:**

- In Table 7, can you provide clarification on why there is a drop in numbers for MGM-SliME-LLaVA-7B ? Overall, why is the performance gain of 3 Models integration is not much compared to the 2 models ensemble ?

---

### Note · Authors · 2024-11-15

I have read and agree with the venue's withdrawal policy on behalf of myself and my co-authors.